# Myrcene Attenuates Renal Inflammation and Oxidative Stress in the Adrenalectomized Rat Model

**DOI:** 10.3390/molecules25194492

**Published:** 2020-09-30

**Authors:** Azim Ullah Shamsul Islam, Björn Hellman, Fred Nyberg, Naheed Amir, Richard L. Jayaraj, Georg Petroianu, Abdu Adem

**Affiliations:** 1Department of Pharmacology and therapeutics, College of Medicine and Health Sciences, United Arab Emirates University, 17666 Al Ain, UAE; azim.sheikh@uaeu.ac.ae (A.U.S.I.); naheed_amir@uaeu.ac.ae (N.A.); richardlj@uaeu.ac.ae (R.L.J.); 2Department of Pharmaceutical Biosciences, Biomedical Centre, Uppsala University, 591751 24 Uppsala, Sweden; Bjorn.Hellman@farmbio.uu.se (B.H.); Fred.Nyberg@farmbio.uu.se (F.N.); 3Department of Pharmacology and Therapeutics, College of Medicine and Health Sciences, Khalifa University, 127788 Abu Dhabi, UAE; Georg.Petroianu@ku.ac.ae

**Keywords:** glucocorticoids, adrenalectomy, inflammation, oxidative stress, Th1 and Th2 cytokines, kidney injury molecule

## Abstract

Physiological Glucocorticoids are important regulators of the immune system. Pharmacological GCs are in widespread use to treat inflammatory diseases. Adrenalectomy (ADX) has been shown to exacerbate renal injury through inflammation and oxidative stress that results in renal impairment due to depletion of GCs. In this study, the effect of myrcene to attenuate renal inflammation and oxidative stress was evaluated in the adrenalectomized rat model. Rats were adrenalectomized bilaterally or the adrenals were not removed after surgery (sham). Myrcene (50 mg/kg body weight, orally) was administered post ADX. Myrcene treatment resulted in significant downregulation of pro-inflammatory cytokines (IL-1β, IL-6, and TNF-α) compared to untreated ADX rats. In addition, myrcene resulted in significant downregulation of immunomodulatory factors (IFNγ and NF-κB) and anti-inflammatory markers (IL-4 and IL-10) in treated ADX compared to untreated ADX. Myrcene significantly increased the antioxidant molecules (CAT, GSH, and SOD) and decreased MDA levels in treated ADX compared to untreated. Moreover, myrcene treatment reduced the expression of COX-2, iNOS, KIM-1, and kidney functional molecules (UREA, LDH, total protein, and creatinine) in ADX treated compared to ADX untreated. These results suggest that myrcene could be further developed as a therapeutic drug for treatment of kidney inflammation and injury.

## 1. Introduction

Glucocorticoids hereafter (GCs) are important regulators of the immune system, while pharmacological GCs are in widespread use to treat inflammatory diseases. The major GC in rodents is corticosterone, which is synthesized in the adrenal gland in response to stress or systemic cytokine release [1]. Adrenalectomy hereafter (ADX) has been shown to exacerbate clinical disease and experimental autoimmune encephalomyelitis. Glomerulonephritis (GN) in humans is modulated by pharmacologic doses of steroids, but the role of endogenous GC in the modulation of glomerular injury has not been revealed. Adrenalectomy results in neutrophil influx, which is required for development of renal injury [2]. There has been renewed interest in physiological adrenal GCs and their role in regulating in vivo immune responses. While pharmacological doses are generally immunosuppressive, they may not accurately describe the effects of physiological levels of endogenous GCs on in vivo immune function. Several in vivo processes related to T cell function are facilitated by GC regulation of cell trafficking in kidney. In addition, GCs are implicated in the regulation of Th1/Th2 balance, such that GCs in vivo can reduce Th1 cytokines (IFNγ and IL-12) and increase Th2 cytokines (IL-4 and IL-10) [3]. In rats, in the second postnatal week, plasma glucocorticoids are increased, and elevated glucocorticoid in this period leads to hypertension in the adult rat and inhibits proliferation of Henle’s loop and the kidney outer medulla. COX-2, a regulator of inflammation, and COX-2 blocker impaired the concentrating ability of kidney in adrenalectomized rats [4,5]. GCs modulate the expression of cytokines, adhesion molecules, chemoattractants, and other inflammatory mediators and molecules and affect immune cell trafficking, migration, maturation, and differentiation. GCs exert immunomodulatory effects through nuclear factor-κB hereafter (NF-κB), which mediates pro-inflammatory actions [6]. The anti-inflammatory actions of GCs reduce cytokines and pro-inflammatory transcription factors such as NF-κB in different pathways, by which they can provide protection against renal inflammation. Exercise and stress elevate circulating GCs and reactive oxygen species (ROS) generation, contributing to exercise-induced lymphocyte apoptosis. ADX does not inhibit exercise-associated intestinal lymphocytes cell loss and administration of the antioxidant to ADX mice protects them from apoptotic cell death [7]. ADX enhances the inhibitory effect of TNF-α on various hormones under stress condition [8]. Inflammation in kidneys reduces clearance, and reduced clearance is also expected to proportionally increase the drugs and toxic molecules in the kidney as well as renal side effects. Inflammation results in downregulation of kidney function [4]. Adrenal insufficiency is an uncommon cause of hypercalcemia and shows that adrenal insufficiency may manifest as hypercalcemia and acute kidney injury, which implicates adrenal insufficiency as a cause of hypercalcemia in clinical practice [9]. The standard therapy in patients with lupus nephritis now includes a combination of GCs with immunosuppressive drugs, which can improve renal outcomes better than either alone. Regardless of whether innate or adaptive immunity is involved, or whether the renal disease is acute or chronic, it is clear that inflammatory cytokines have a central role as both mediators of immune function and initiators of renal injury. However, cytokines have immunomodulatory roles that can abrogate the development of renal disease [10]. Kidney injury molecule-1 hereafter (KIM-1) is upregulated in animal proteinuric renal disease. Elevated KIM-1 could be quantified and related to the extent of renal damage. KIM-1 is related to renal KIM-1 expression, the degree of renal morphological damage, renal function, and proteinuria, which is associated with tubulointerstitial damage and inflammation. KIM-1 is a marker for renal proximal tubular damage, a potential biomarker of renal damage, and a predictor of renal function [11].

The release of endogenous GCs is essential for preventing exaggerated immune responses, whereas adrenalectomy results in high mortality in various models. In the case of macrophages and the expression of pro-inflammatory cytokines, GCs repressed inducible nitric oxide synthase hereafter (iNOS). Consequently, production of oxygen radicals is reduced and phagocytotic activity impaired. Corticosterone modulates macrophage function dependent on its concentration [1]. It is known that GCs prevent the induction of iNOS in rats. Inhibition of iNOS protects sham rats against lethal shock, while the absence of GCs may increase the iNOS gene expression, with NO-overproduction [12]. Adrenal insufficiency lowers basal cortisol levels of GCs, increases prooxidant biomarker (MDA), and decreases antioxidant biomarkers CAT and SOD [9,12].

Myrcene is recognized for many years as vital anti-inflammatory agent in various disease. Martin et al. demonstrated that myrcene possesses antioxidant [13,14] and anti-inflammatory [15,16,17] properties as well as prevents cytotoxicity. Myrcene significantly inhibited NO production stimulated by LPS in macrophages, indicating it has anti-inflammatory properties. Myrcene downregulated the production of NO, reduced NF-κB and iNOS [18] expression induced by IL-1β, and inhibited iNOS [17]. The mechanism of myrcene to inhibit the production of oxidative free radicals is direct action on target cells; however, in analgesia, its action is via transient receptor potential cation channel subfamily V member 1 (TRPV1) [19,20].

## 2. Results

### 2.1. Effect of ADX on Body Weight

The body weight of ADX rats significantly (*p* < 0.05) reduced from 165 ± 9.9 to 136 ± 11.20 g over the two weeks of ADX, while the weight of the sham operated increased significantly (*p* < 0.01) from 165.67 ± 8.45 to 202 ± 8.94 g.

### 2.2. Effect of Myrcene on Inflammatory Cytokines, Immune Regulatory and Anti-Inflammatory Cytokines in the Kidneys after ADX

The concentrations of pro-inflammatory cytokines IL-1β, IL-6, TNF-α, and NF-kB/p65; immune regulatory cytokine IFN-γ; and anti-inflammatory cytokines IL-4 and IL-10 were measured in kidney homogenates of 14 days adrenalectomized rats treated or untreated with myrcene compared to sham operated group, respectively (Figure 1). In myrcene untreated ADX rat kidney homogenates, IL-6, IL-1β, and TNF-α concentrations were significantly elevated as compared to naïve and sham untreated control group, while treatment with myrcene significantly downregulated the concentration of these cytokines in kidney homogenates as compared to the untreated ADX group and were found to be at the same level of sham operated group (Figure 1).

### 2.3. Effect of Myrcene on Immunomodulatory and Anti-Inflammatory Cytokines after ADX

The immune regulatory cytokine IFNγ concentration in the kidney was found to be significantly elevated in the ADX group compared to the naïve and sham operated group; however, there was no change in the IFNγ concentration in the myrcene treated ADX group compared to the untreated ADX control group (Figure 2A).

The anti-inflammatory cytokines IL4 and IL-10 were measured on Day 14 after adrenalectomy in the kidneys of myrcene treated and untreated ADX rats as well as sham operated rats. IL-4 concentration was significantly increased in the ADX untreated controls as compared to the naïve and untreated sham control group, while IL-10 was significantly decreased in the untreated ADX controls as compared to naive control group. The myrcene treatment of the ADX rats resulted in significantly lower IL-4 and significantly higher IL-10 compared to the untreated ADX control group (Figure 2B,C).

### 2.4. Effect of Myrcene on Lipid Peroxidation and Glutathione Level in Kidneys of ADX Animals

To investigate the effect of myrcene on oxidative kidney damage due to adrenalectomy, we measured lipid peroxidation and glutathione level (Figure 3). The lipid peroxidation (MDA) was significantly increased while the GSH level was significantly decreased in the untreated ADX control rats compared to the naïve and sham control group. Myrcene treatment of the ADX rats significantly decreased the lipid peroxidation and significantly increased the glutathione level compared to the untreated ADX control rats.

### 2.5. Effect of Myrcene on SOD and CAT Antioxidant Enzymes after ADX

The SOD and CAT enzymes were measured in kidneys of naïve, untreated and myrcene treated ADX and sham operated rats. SOD and CAT levels were significantly lower in untreated ADX control group compared to untreated control sham operated group. Myrcene treatment significantly upregulated SOD and CAT enzymes compared to the untreated ADX rats (Figure 3C,D).

### 2.6. Effect of Myrcene on KIM-1 after ADX

KIM-1 was measured at 14 days after adrenalectomy in kidney homogenates. KIM-1 was significantly increased in untreated ADX controls compared to sham operated controls. Myrcene treatment resulted in significantly decreased KIM-1 in treated ADX compared to untreated ADX controls (Figure 4).

### 2.7. Effect of Myrcene on Kidney Function after ADX

The kidney function molecules LDH, BUN, creatinine, and total protein were measured in serum. The untreated ADX group showed significantly higher concentrations of LDH, BUN, and creatinine compared to untreated sham operated group, indicating impaired kidney function. No significant difference was observed for the total protein (Figure 5). Myrcene treatment of the ADX rats significantly decreased the LDH, BUN, and creatinine concentrations compared to the untreated ADX control group (Figure 5).

### 2.8. Effect of Myrcene on the Expression of Inflammatory Mediators: COX-2 and iNOS

At 14 days post adrenalectomy, we investigated the expression of COX-2 and iNOS using Western blots in kidney tissue lysates (Figure 6). A 1.4-fold increase in COX-2 expression was observed in the untreated control ADX compared to naïve and untreated sham operated control group. However, myrcene treatment reduced the COX-2 expression by 1.3 folds compared to the untreated ADX rats. iNOS expression was 1.3-fold increase in untreated ADX group compared to naïve and sham operated group. The myrcene treated ADX rats showed a 1.8-fold iNOS reduction compared to untreated ADX group.

## 3. Materials and Methods

### 3.1. Animals and Experimental Design

Male Wistar albino rats were purchased from Harlan Laboratories (Oxon, UK). The animals were bred in Animal Facility (CMHS) UAE University, Al Ain, United Arab Emirates. Rats weighing 170 ± 10 g were used. Rats were anesthetized intraperitoneally with ketamine–xylazine Cat: K113, mixture (ketamine 100 mg/kg Cat: K101 body weight sigma, St. louis, MO, USA, and xylazine 5 mg/kg body weight, sigma, St. louis, MO, USA. Rats were adrenalectomized bilaterally with open surgical procedure. Sham operated or naive rats were used as controls. ADX rats were kept with 0.9% NaCl in the drinking water to restore body electrolytes. Myrcene treated group was given 100 mg/kg body weight post ADX for 14 days. Rats were decapitated, and blood and kidney were collected. The blood was processed to collect sera. The kidney samples (100 mg/mL) were incubated in KCl buffer (Tris-HCl 10 mM, NaCl 140 mM, KCl 300 mM, EDTA 1 mM, Triton X-100 0.5%, (prepared in lab) at pH 8.0 supplemented with protease and phosphatase inhibitor homogenized with Ultra Taurrax T-25 (Janke and Kunkel, IKA laboratories, (Staufen. Germany) and the supernatants were collected after centrifugation at 13,000 rpm for 30 min at 4 °C.

### 3.2. Glutathione (GSH) Assay

GSH content in kidney homogenate was estimated according to the method described by sigma assay kit. The measurement of GSH uses a kinetic assay in which catalytic amounts (nmol) of GSH cause a continuous reduction of 5,5-dithiobis (2-nitrobenzoic acid) to nitrobenzoic acid (TNB) and the glutathione disulfide (GSSG) formed is recycled by glutathione reductase and NADPH. The yellow color product, 5-thio-2-TNB, was measured spectrophotometrically at 412 nm within 5 min of 5,5-dithio-bis (2-nitrobenzoic acid) addition, against a blank with no homogenate. GSH concentration was expressed as μM of GSH.

### 3.3. ELISA of Cytokines

ELISA Kits for IL-1b, IL-6, TNF-α, IFN-γ, IL-4, and IL-10 were purchased from R&D systems (Minneapolis, MN, USA), and ELISA was performed according to the kits instruction. Briefly, 96-well plates (Thermo scientific, product number # 439454, Waltham, MA, USA) were coated overnight with 100 µL of primary antibody (Ab) in phosphate buffer saline (PBS), (Sigma, St. louis, MO, USA). The ELISA plates were washed three times with washing buffer (PBS containing 0.05% Tween 20). The plates were blocked with 1% bovine serum albumin for 1 h. One hundred microliters of standard or samples were added to the wells and incubated for 2 h at room temperature. The plates were washed three times and then dispensed with detection Ab in for 2 h. One hundred microliters of HRP were added for 30 min. Color was developed with tetra methyl benzidine (TMB, (R&D systems Minneapolis, MN, USA) in dark until optimal reaction period, and then the reaction was terminated with stop solution (2 N H_2_SO_4_). The ELISA plates were read for absorbance value (OD) with ELISA reader Tcan provided with Magellan software, (Mannedorf, Switerzland) at 450-nm wavelength with 620 nm as reference wavelength. Tissue culture supernatants and kidney homogenates content were determined for IL-6, IL-1β, TNF-α, IFN-γ, IL-4, and IL-10 using a Duoset ELISA development kit from according to the manufacturer’s instructions.

### 3.4. Estimation of NF-κB in Kidney Homogenate

NF-κB/p65 Active NBP2-29661 NOVUS was used following the kit protocol. Briefly, the ELISA plate was coated with 100 µL of primary antibody for overnight. The next day, the ELISA plate was washed two times and then blocked with 200 µL of blocking buffer for 1 h. Standard concentrations of positive control, negative control, samples. and blank control were dispensed with 100 µL and incubated overnight at 4 °C. Diluted secondary antibody in a volume of 100 μL was dispensed to each well and incubates for 1 h at RT. The ELISA plate was washed four times. The substrate was dispensed and incubated for 30 min for optimal color development. The ELISA plates were read at 405 nm and the concentration was calculated.

### 3.5. Estimation of the Activities of Enzymatic Antioxidants

Cayman assay kits (Cayman Chemicals Company, Ann Arbor, MI, USA) were used to determine the activities of antioxidant enzymes such as superoxide dismutase (SOD) and catalase (CAT) following manufacturer’s instructions. Briefly, CAT was estimated by adding the 20-µL samples or standards of different concentration to 100 µL of assay buffer and 30 µL of methanol in 96-well plates. Twenty microliters of H_2_O_2_ were added to initiate the reaction with 20 min of incubation at room temperature (RT). Thirty microliters of potassium hydroxide were added to terminate the reaction and subsequently 30 µL of catalase purpald (chromogen) and 10 µL of catalase potassium periodate were added. The plate was incubated for 5 min at RT on a shaker and the absorbance was read at 540 nm by using the microplate reader.

For SOD measurement, 10-µL samples or standard were added to 96-well plates. Xanthine oxidase (20 µL) was added to each well to initiate the reaction. The plate was shaken for a few seconds and then covered with plate cover and incubated for 30 min at RT. Absorbance was read at 450 nm by using the microplate reader. The CAT activity was expressed as nmol/min/mg protein and the SOD activity was expressed as units/mg protein.

### 3.6. Estimation of Kidney Function Test

The blood was centrifuged at room temperature for 15 min at 3000 rpm, and the serum samples were stored at −80 °C until further analysis of the activities of lactate dehydrogenase (LDH). Urea and creatinine were measured by standard laboratory methods using commercially available kits (Hoffman-La Roche Ltd., Basel, Switzerland) and a LX20 multiple automated analyzer (Beckman Coulter, Brea, CA, USA).

### 3.7. Western Blot Quantification of COX-2 and iNOS Expression

Western blot quantification was measured to analyze COX-2 (abcam, Cambridge, MA, USA) and iNOS (sigma, St. louis, MO, USA) expression. Kidney tissues isolated from each animal were homogenized in KCL buffer with protease and phosphatase inhibitors. The cell lysates were centrifuged at 13,000 rpm for 30 min. The supernatant containing cytoplasmic fractions was isolated, and protein concentration was measured as described below. The cytoplasmic fraction containing equal amounts of protein (35 μg) were loaded and separated using 10% SDS–polyacrylamide gel electrophoresis. The proteins were then transferred onto a PVDF membrane using Trans-Blot Turbo Transfer System, (Bio-rad, Hercules, CA, USA). The membranes were washed and blocked using 5% skimmed milk. Next, the membranes were incubated overnight at 4 °C with specific primary rabbit polyclonal antibodies against COX-2 (1:1000) and iNOS (1:5000). The next day, the membranes were washed and then incubated with horseradish peroxidase-conjugated secondary anti-rabbit antibody. The protein recognized by the antibody was visualized using an enhanced chemiluminescence Pico kit (Thermo Fisher Scientific, Rockford, IL, USA). The blots were stripped and re-probed for β-actin (1:5000; monoclonal mouse, Millipore, St. louis, MA, USA) as a loading control. The intensity of the bands was measured using densitometry and quantified using Image J software (NIH, Bethesda, Rockville, MD, USA). Not applicable, download form google

### 3.8. Protein Estimation

The total protein was estimated using the Pierce BCA protein assay kit (Thermo Fisher Scientific) following the manufacturer’s instructions.

### 3.9. Statistical Analysis

The experimental data analysis was carried out with SPSS software version 24, (IBM Middle East, Dubai, UAE) using student’s t-test. Data were expressed as mean ± STD.

## 4. Discussion

GCs are important regulators of the immune system. Physiological GCs are pivotal to maintain normal immune system, while pharmacological GCs are in use to suppress the immune system, particularly inflammatory factors [1]. Removal of adrenal glands (ADX) results in the depletion of glucocorticoids. Depletion of GCs (ADX) results in the influx of immune cells, particularly macrophages and NK cells in kidneys, which leads to kidney diseases (glomerulonephritis) and other kidney complications [2]. Kidney diseases are mainly acute kidney injury or chronic kidney disease, including oxidative stress and inflammation. Inflammation plays a major role in the pathophysiology of chronic kidney diseases. Inflammatory cytokines (IL-6, IL-1β, and TNF-α) have a central role as mediators of immune function and initiators of renal injury [10]. The potential markers involved in kidney injury are pro-inflammatory cytokines, impaired regulation of immune regulatory cytokines (IFNγ), anti-inflammatory cytokines (IL-4 and IL-10), and ROS [21]. Various reports demonstrated that myrcene has potential anti-inflammatory and antioxidant properties [13,14,15,16,17,18,19,20]; however, its anti-inflammatory and antioxidant properties in ADX are yet to be determined. Inflammatory cytokines were measured to determine the anti-inflammatory role of myrcene on kidneys of myrcene treated and untreated ADX rats. Our data show that pro-inflammatory cytokines (IL-6, IL-1β, and TNF-α) were increased in untreated ADX group as compared to sham control. In addition, IL-6, IL-1β, and TNF-α were downregulated in myrcene treated ADX group when compared to ADX untreated group indicating the anti-inflammatory properties of myrcene. The macrophage activation requires IFN-γ and hence increase the pro-inflammatory cytokines [22]. IL-12 induced IFN-γ and its sustained presence results in kidney injury. Schwarting et al. (1999) showed that INF-γ either promotes or limits renal injury depending on acute or chronic phase [23]. In acute infection, IFN-γ is needed to activate the macrophages. However, in sustained inflammation, IFN-γ is needed to downregulate the activation of macrophages and results in lowering the inflammatory cytokines. Another study showed that the presence of IFN-γ is required to suppress the progression of renal injury in chronic inflammation [24]. Pro-inflammatory cytokines are involved in the pathogenesis of renal disease, by upregulating endothelial cell adhesion molecules and chemokines that further promote renal immune cell infiltration [24,25,26]. Our data show a significant increase in IFN-γ in the untreated ADX group compared to sham control while treatment with myrcene further increased IFN-γ significantly. This significant increase of IFN-γ in the myrcene treated group would mitigate the inflammation in treated ADX group. Pro-inflammatory cytokines activate NF-κB and the activation of NF-κB is parallel with the pro-inflammatory cytokines. This activation is due to the activation of B cells (NF-κB) [27,28]. NF-κB expression is increased in kidneys associated with glomerulonephritis [29]. Hence, expression of NF-κB was measured 14 days post ADX. The results show that NF-κB was upregulated in the untreated control ADX compared to sham control while myrcene significantly lowered NF-κB concentration in the treated ADX group. Our results show that myrcene resulted in downregulation of pro-inflammatory cytokines as well as NF-κB, activator of B cells, leading to adaptive immunity.

Th cells are classified into subsets based largely on the type of cytokine they produce or their primary function and include Th1, Th2 [30], and T regulatory cells. Th1 cells are characterized by the production of inflammatory cytokines including IFN-γ and TNF-α. The Th1 cells promote the activation of macrophages that drive tissue injury [31]. Th2-polarized cells primarily secrete anti-inflammatory cytokines (IL-4 and IL-10) that can downregulate macrophage activation [30]. Cytokines from Th2 polarized cells have multiple roles in immune regulation. The differing functions of Th1 and Th2 cytokines ultimately led to a proposed model for describing renal diseases as either Th1 or Th2 dominated [3,31]. Depletion of glucocorticoids (ADX) leads to lethal inflammatory reaction to endotoxin [32,33], which is prevented by dexamethasone [34,35]. Our results show significantly elevated IL-4 and significantly reduced IL-10 in untreated ADX rats compared to sham controls. IL-10 is a potent anti-inflammatory cytokine that plays a crucial, and often essential, role in preventing inflammation. IL-10 was impaired in untreated ADX group while myrcene treatment resulted in significant IL-10 upregulation as well as downregulation of IL-4, suggesting myrcene as an anti-inflammatory. IL-10 downregulates the expression of Th1 cells and hence results in anti-inflammatory effect and decrease in macrophage activation. In addition, IL-10 decreases NF-κB activity.

It is known that cyclooxygenase-2 (COX-2) is a regulator of inflammation, and corticosterone depletion (ADX) elevates the expression of COX-2, resulting in impaired urinary concentrating ability in adrenalectomized rats [3,36]. Inflammation reduces clearance in kidney, and reduced urinary clearance increases the accumulation of toxic drugs and molecules in the kidney. Inflammation influences the distribution of COX-2 inhibitors in the kidney and results in lower kidney function. The systemic signs of inflammation such as increased pro-inflammatory cytokines result in upregulation of inflammation in the kidneys, which ultimately show kidney damage [3]. Cox-2 was significantly elevated in untreated ADX group compared to the sham group, showing impaired control of inflammation and postulates kidney dysfunction. Conversely, treatment with myrcene resulted in downregulation of Cox-2 in the ADX treated group.

KIM-1 elevation is a sign of inflammation and has been shown to be increased in acute and chronic kidney injury [11]. KIM-1 is believed to play a role in tubulo-interstitial damage [37]. KIM-1 is non-detectable in normal kidneys, but strong KIM-1 induction has been shown in proteinuric renal disease [38]. Tubular KIM-1 expression is specific to ongoing tubular cell damage and is strongly induced in acute and chronic kidney inflammation [11,39,40,41]. Our data show a significant increase in KIM-1 levels in the untreated ADX group, indicating tubular cell damage. However, myrcene treatment resulted in downregulation of KIM-1 signifying the importance of myrcene as a protective agent in renal disease.

Oxidative stress is defined as the imbalance of reactive oxygen species (ROS) produced within the cells and the endogenous cellular potential to neutralize these harmful intermediates. GCs and reactive oxygen species generation are increased in exercise and adrenalectomy (ADX) does not inhibit exercise-associated cell loss during exercise [7]. It is well known that glucocorticoids prevent the induction of iNOS. GCs deficiency results in overexpression of iNOS and causes oxidative damage in cells. The over expression of iNOS leads to over production of nitric oxide and the production of free radicals causes tissue damage [12]. Similarly, another study showed that adrenal insufficiency significantly enhanced endotoxemia-induced iNOS expression, and nitric oxide (NO) formation caused oxidative stress in the adrenal gland, as evidenced by the increase of lipid peroxidation biomarker (MDA) and the decrease of antioxidant biomarkers (CAT and SOD activity [9]. Inducible nitric oxide synthase was highly expressed in the untreated ADX group compared to sham, indicating enhanced ROS molecules. However, ADX and sham treated with myrcene showed decrease in iNOS and ROS molecules. Sham treated and non-treated controls showed no significance difference depicting intact oxidative system. GSH, CAT, and MDA evaluation showed the antioxidant status in kidneys of the untreated ADX rats. It was found that GSH and CAT were significantly lower in the untreated ADX than sham control. The kidney in the untreated ADX group was therefore vulnerable to free oxygen radicals. MDA was significantly higher in the untreated ADX than sham, explaining the role of free oxygen radicals in oxidative damage of the kidneys. Myrcene treatment resulted in significant upregulation of antioxidant enzymes GSH and CAT and downregulated MDA level, suggesting an antioxidant role for myrcene in renal diseases.

Inflammatory cytokines (IL-6, IL-1β, and TNF-α) have a central role in mediation of renal injury due to their continuous presence [10]. Various pathological factors such as IFNγ [28], NF-κB [29,30], and KIM-1 play a role in tubulo-interstitial damage in kidney [42]. As tubular KIM-1 expression is specific to ongoing tubular cell damage and strongly induced in acute and chronic kidney inflammation, iNOS and reactive oxygen species lead to kidney damage [14,15,35]. Inflammation contributes to maintaining fluid balance and removal of uremic toxins. Hence, this results in increases of blood urea nitrogen (BUN), creatinine, and other molecules [43]. BUN, LDH, and creatinine levels were significantly higher in serum of untreated ADX group as compared to sham controls, indicating impairment of kidney function, while myrcene treatment reversed the serum BUN and creatinine level to the normal baseline. However, the LDH level was significantly downregulated in the myrcene treated ADX rat but still did not return to baseline and was significantly higher than sham control group. The present study showed that myrcene resulted in downregulation of inflammation, oxidative damage, and KIM-1, which resulted in improved kidney function. Myrcene treatment resulted in downregulation of pro-inflammatory cytokines, immune modulatory cytokine, KIM-1, NF-κB, and increase in anti-inflammatory cytokines. These results suggest that myrcene could be further developed as a therapeutic drug for treatment of kidney inflammation and injury.

## Figures and Tables

**Figure 1 molecules-25-04492-f001:**
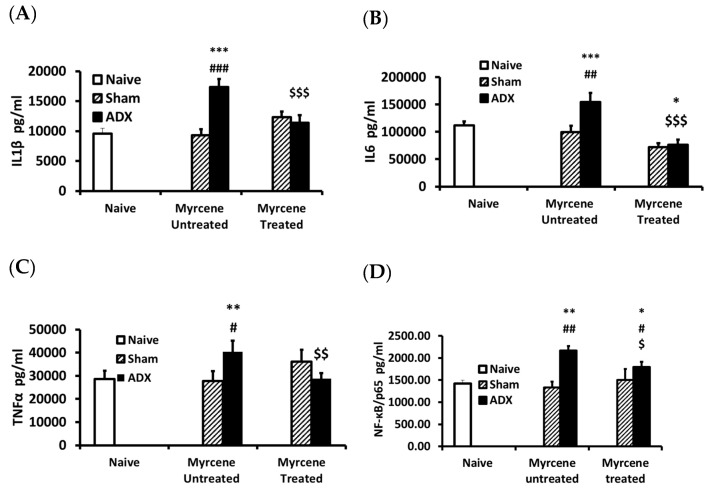
Pro-inflammatory cytokines levels in kidneys of ADX, sham operated, and myrcene treated sham and ADX rats: IL-1β (**A**); IL-6 (**B**); TNF-α (**C**); and NF-қB (**D**). The ADX group showed significantly increased concentrations of IL-1β, IL-6, TNFα, and NF-қB compared to sham operated group. Myrcene decreased significantly IL-1β, IL-6, TNF-α, and NF-қB compared to untreated ADX, while sham controls (untreated and treated) remained unchanged. Each data-point represents the mean ± STD of 6–9 rats per group. * *p* < 0.05, ** *p* < 0.01, *** *p* <0.001 compared to naïve; ^#^
*p* < 0.05, ^##^
*p* < 0.01 ^###^
*p* < 0.001 compared to untreated sham; ^$^
*p* < 0.05, ^$$^
*p* < 0.01, ^$$$^
*p* < 0.001 compared to untreated ADX.

**Figure 2 molecules-25-04492-f002:**
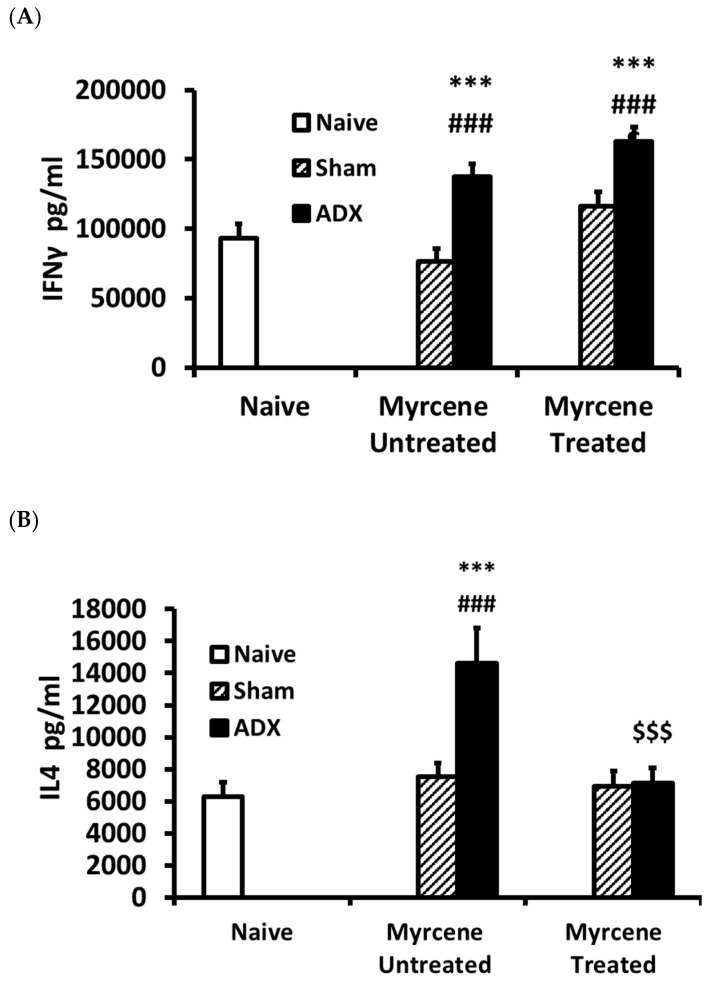
Immune modulatory and anti-inflammatory cytokines levels in ADX, sham operated, and myrcene treated ADX and sham operated rats: IFN-γ (**A**); IL-4 (**B**); and IL-10 (**C**). In the untreated ADX group, IFN-γ and IL-4 were upregulated significantly compared to sham operated group. Myrcene did not decrease IFNγ, while significantly decreased IL-4 compared to the untreated ADX. In the untreated ADX group, IL-10 concentration was significantly lower than sham controls, while myrcene treatment significantly upregulated IL-10 compared to the untreated ADX group. * *p* < 0.05, ** *p* < 0.01, *** *p* < 0.001, compared to naïve; ^##^
*p* < 0.01 ^###^
*p* < 0.001 compared to untreated sham; ^$$^
*p* < 0.01, ^$$$^
*p* < 0.001 compared to untreated ADX.

**Figure 3 molecules-25-04492-f003:**
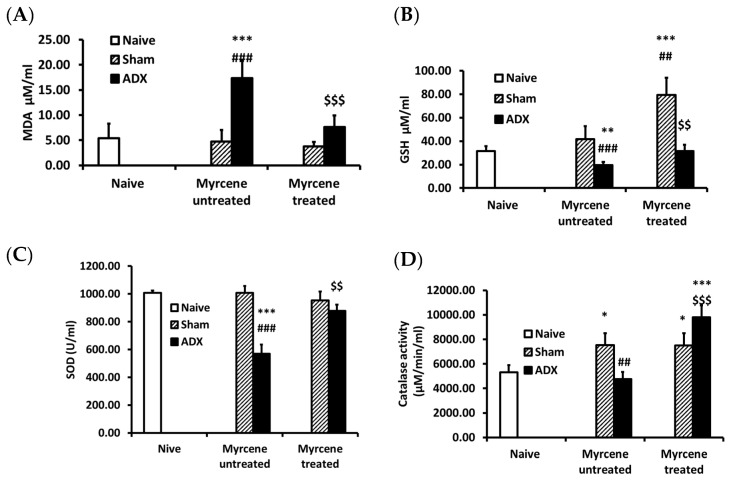
Quantification in kidney tissue after adrenalectomy of: (**A**) malondialdehyde (MDA); (**B**) glutathione (GSH); (**C**) superoxide dismutase (SOD); and (**D**) catalase (CAT). ADX induced a significant increase in MDA level and decreased GSH level in the untreated ADX group compared to the naïve and sham operated control rats. Myrcene treatment in ADX group rats significantly decreased the concentration of MDA, and increased the level of GSH. ADX significantly decreased the activity of SOD and CAT compared to the naïve and sham control rats. Myrcene treatment significantly up regulated SOD and CAT activity compared to the untreated ADX rats. Each data-point represents the mean ± STD of six to nine rats. * *p* < 0, ** *p* < 0.01, *** *p* < 0.001 compared to naïve; ^##^
*p* < 0.01, ^###^
*p* < 0.001 compared to untreated sham; ^$$^
*p* < 0.01, ^$$$^
*p* < 0.001 compared to untreated ADX.

**Figure 4 molecules-25-04492-f004:**
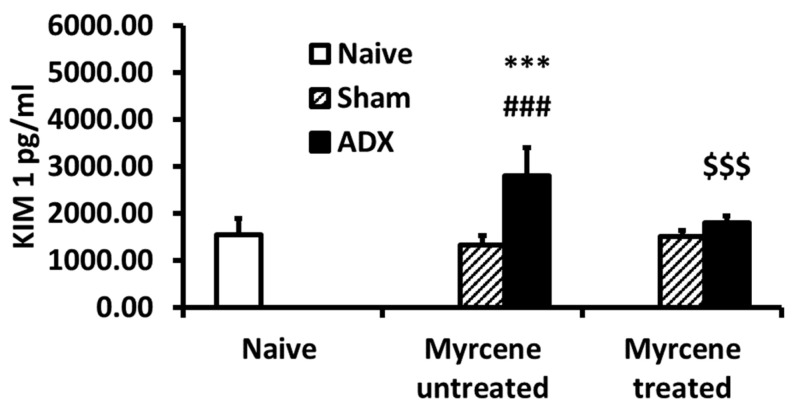
Quantification of Kidney KIM-1 in kidney tissue of naïve, untreated and myrcene treated ADX rats. ADX induced a significant increase in KIM-1 in the untreated ADX rats compared to the naïve and sham operated control rats. Myrcene treatment of ADX group rats significantly decreased the concentration of KIM-1. Values are expressed as the mean ± SEM (*n* = 6–8). *** *p* < 0.001 compared to naïve; ^###^
*p* < 0.001 compared to sham untreated; ^$$$^
*p* < 0.001 compared to untreated ADX.

**Figure 5 molecules-25-04492-f005:**
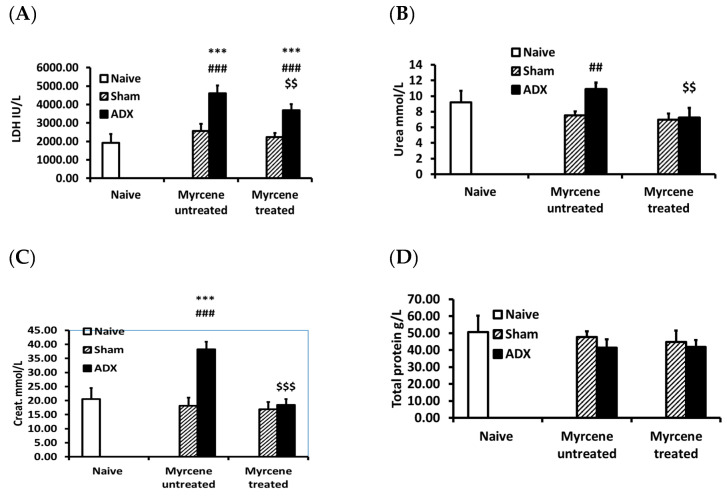
Serum kidney function test molecules in ADX, sham operated rats and myrcene treated ADX and sham rats: (**A**) LDH; (**B**) BUN; (**C**) creatinine; and (**D**) total protein. LDH, BUN, and creatinine were significantly increased in the untreated ADX group compared to the sham operated. Myrcene treatment significantly decreased LDH, BUN, and creatinine compared to untreated ADX group. However, the LDH levels of the myrcene treated ADX group were significantly upregulated compared to sham treated group. Each data-point represents the mean ± STD of 6–9 rats per group. *** *p* < 0.001 compared to naïve; ^##^*p* < 0.01, ^###^
*p* < 0.001 compared to untreated sham; ^$$^
*p* < 0.01, ^$$$^
*p* < 0.001 compared to untreated ADX.

**Figure 6 molecules-25-04492-f006:**
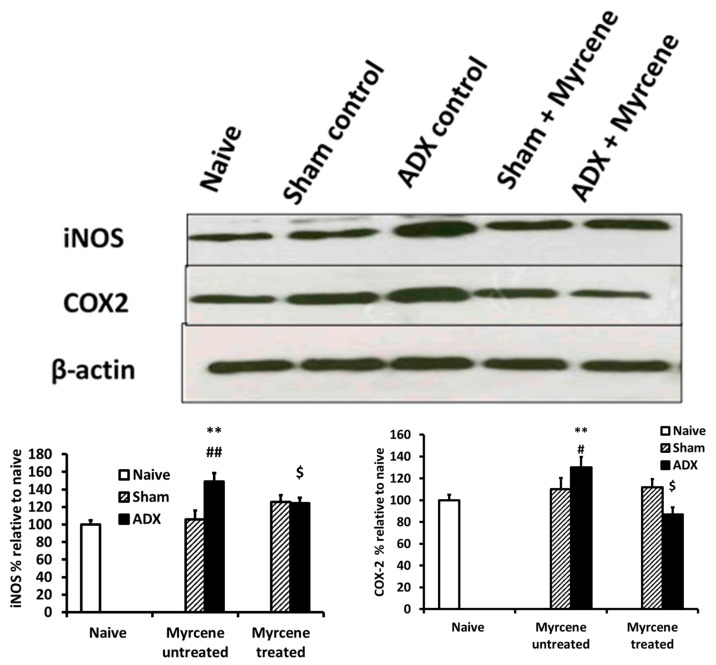
Quantification of iNOS and COX-2 in kidney tissue of ADX, naïve, sham operated, and myrcene treated ADX and sham rats measured by western blotting. A significant increase in iNOS and COX-2 was observed in the untreated ADX group rats compared to the naïve and sham operated control. Myrcene treatment significantly decreased the expression of iNOS and COX-2. Values are expressed as the mean ± STD (*n* = 3). ** *p* < 0.01 compared to naïve; ^#^
*p* < 0.05, ^##^
*p* < 0.01 compared to untreated sham; ^$^
*p* < 0.05 compared to untreated ADX.

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
