# Peer review of "Myrcene Attenuates Renal Inflammation and Oxidative Stress in the Adrenalectomized Rat Model"

_molecules, 2020, doi:10.3390/molecules25194492_

Round 1

Reviewer 1 Report

Myrcene also showed powerful anti-inflammatory and anticatabolic effects in a human chondrocyte model of osteoarthritis. Authors have shown the its effect on the inflammation development in kidenys. The paper is well-daon and written in clear scientific style.

Author Response

We thank reviewer 1 for his time and revision.

Reviewer 2 Report

Major points

  1. Discussion section should be supported by new literature. Please include a critical discussion about the possible mechanism of myrcene in this model at molecular level. The relationship or crosstalk between inflammation and oxidative stress in this model should be explained. How myrcene attenuates inflammation and oxidative stress? It should be explained if myrcene is a direct, indirect or bifunctional antioxidant

Minor points

  1. The abbreviations should be defined in parentheses the first time they appear be used consistently thereafter. For example the abbreviations such as glucocorticoids (GCs), adrenalectomy (ADX), kidney injury molecule-1 (KIM-1) and reactive oxygen species (ROS) to mention some of them, are not used consistently in the abstract, main text and discussion. In addition, some abbreviations are not defined and are used in the manuscript.
  2. Introduction section should be supported by new literature and must be improved, is necessary summarized information related with glucocorticoids, added more information about myrcene in renal models and mention the aim of this study.
  3. There are missing or extra spaces in the manuscript.
  4. Lines 43 and 45. Please change the words “in vivo” to cursive letter.
  5. Line 98. “gram” should be “g”.
  6. Line 106. “inhibitor.homogenized” should be “inhibitor homogenized”.
  7. Line 108. “4 °c” should be “4°C”.
  8. Line 110. “GSH content in serum and heart homogenate was estimated” however, in Figure 3 (line 235) the results are in kidney homogenate. Please clarify the respective section.
  9. Lines 114 and 127. “412” and “620” should be “412 nm” and “620 nm”.
  10. 131-137. “Estimation of NF-kB is kidney homogenate”, “ul” and “hr” should be “Estimation of NF-kB in kidney homogenate”, “ml” and “h”, respectively.
  11. Line 154. The units of the expressions of results of CAT and SOD should be defined.
  12. Lines 181-182. Please cite adequately the statistics software used and “student’s t-est” should be “student’s t-test”.
  13. Line 185. “0020reduced from” should be “reduced from”.
  14. Figure 3. The results of Figure 3 should be corrected by g protein.
  15. Figure 5. The Axis title: “Creat. Mmol/L” should be “Creat. mmol/L”.
  16. The references are numbered, however it appears in disorder in the manuscript. In addition, the reference list does not include DOI numbers.

Author Response

Reviewer 2

  1. Discussion section should be supported by new literature. Please include a critical discussion about the possible mechanism of myrcene in this model at molecular level. The relationship or crosstalk between inflammation and oxidative stress in this model should be explained. How myrcene attenuates inflammation and oxidative stress? It should be explained if myrcene is a direct, indirect or bifunctional antioxidant.

        Discussion is added with new references where applicable.

The new references for myrcene are added and the mechanism is explained. Myrcene affects and showed potent anti-oxidant activities by reducing 1,1-diphenyl-2-picryhydrazyl (DPPH) (,2,2'-azinobis-(3-ethylbenzthiazoline-6-sulphonate) (ABTS) (and superoxide anion free radicals or via transient receptor potential cation channel subfamily V member 1 (TRPV1) in analgesic conditions. Refrences are added in the manuscript.

ROS oxidases IκB kinase, and thus leading to the release of NF- κB. NF- κB, being a nuclear factor activates several inflammatory mediators such as ICAM-1 and VCAM-1 along with pro-inflammatory cytokines. NF- κB also activates macrophages and neutrophils, which results in the release of lysosomal and ROS and eventually damage to target cells.

  1. B. Tenório, R. C. Ferreira, F. A. Moura, N. B. Bueno, A. C. M. de Oliveira, and M. O. F. Goulart, “Cross-Talk between Oxidative Stress and Inflammation in Preeclampsia,” Oxid Med Cell Longev, vol. 2019, pp. 8238727, 2019.

Minor points

  1. The abbreviations should be defined in parentheses the first time they appear be used consistently thereafter. For example the abbreviations such as glucocorticoids (GCs), adrenalectomy (ADX), kidney injury molecule-1 (KIM-1) and reactive oxygen species (ROS) to mention some of them, are not used consistently in the abstract, main text and discussion. In addition, some abbreviations are not defined and are used in the manuscript.

    The following abbreviations are used.

 Abbreviation: adrenalectomy (ADX), glucocorticoids (GC), hypothalamus-          pituitary-adrenal axis (HPA), cyclooxygenase-2 (COX-2), reactive oxygen species (ROS), kidney injury molecule-1 (KIM-1), Inducible nitric oxide synthase (iNOS), Nuclear factor-κB (NF-κB), Malondialdehyde (MDA), Nitric oxide (NO), Sodium oxide dismutase (SOD), Catalase (CAT), Glutathione (GSH)

  1. Introduction section should be supported by new literature and must be improved, is necessary summarized information related with glucocorticoids, added more information about myrcene in renal models and mention the aim of this study.

        More references are added.

  1. There are missing or extra spaces in the manuscript.

        Missing and extra spaces are removed

  1. Lines 43 and 45. Please change the words “in vivo” to cursive letter.

        “in vivo” changed to “in vivo”  

  1. Line 98. “gram” should be “g”.

        Changed accordingly.

  1. Line 106. “inhibitor.homogenized” should be “inhibitor homogenized”.

        Changed accordingly

  1. Line 108. “4 °c” should be “4°C”.

       Changed accordingly

  1. Line 110. “GSH content in serum and heart homogenate was estimated” however, in Figure 3 (line 235) the results are in kidney homogenate. Please clarify the respective section.

       “GSH content in serum and heart homogenate was estimated” Changed to              kidney homogenate  

  1. Lines 114 and 127. “412” and “620” should be “412 nm” and “620 nm”.

        Changed accordingly

  1. 131-137. “Estimation of NF-kB is kidney homogenate”, “ul” and “hr” should be “Estimation of NF-kB in kidney homogenate”, “ml” and “h”, respectively.

        Changed accordingly.

  1. Line 154. The units of the expressions of results of CAT and SOD should be defined.

       The units of the expressions of results of CAT and SOD were defined.

  1. Lines 181-182. Please cite adequately the statistics software used and “student’s t-est” should be “student’s t-test”.

         Changed accordingly

  1. Line 185. “0020reduced from” should be “reduced from”.

        0020 deleted and corrected.

  1. Figure 3. The results of Figure 3 should be corrected by g protein.

        The units are according to the kit instruction.

  1. Figure 5. The Axis title: “Creat. Mmol/L” should be “Creat. mmol/L”.

        Creatinine unit is changed, while others units used and calculated as in kits            instructions.

  1. The references are numbered, however it appears in disorder in the manuscript. In addition, the reference list does not include DOI numbers.

        References are revised